# SciGen: a Dataset for Reasoning-Aware Text Generation from Scientific Tables

**Nafise Sadat Moosavi**[1], **Andreas Rücklé**[1,*] **Dan Roth**[2], **Iryna Gurevych**[1]

[1]Ubiquitous Knowledge Processing Lab (UKP Lab)
Department of Computer Science, Technical University of Darmstadt
`https://www.ukp.tu-darmstadt.de`
[2]Department of Computer and Information Science, UPenn
`https://www.seas.upenn.edu/~danroth`

## Abstract

We introduce SciGen, a new challenge dataset consisting of tables from scientific articles and their corresponding descriptions, for the task of reasoning-aware data-to-text generation. Describing scientific tables goes beyond the surface realization of the table content and requires reasoning over table values. The unique properties of SciGen are that (1) tables mostly contain numerical values, and (2) the corresponding descriptions require arithmetic reasoning. SciGen is the first dataset that assesses the arithmetic reasoning capabilities of generation models on complex input structures, such as tables from scientific articles, and thus it opens new avenues for future research in reasoning-aware text generation and evaluation. The core part of SciGen, including the test data, is annotated by one of the authors of the corresponding articles. Such expert annotations do not scale to large training data sizes. To tackle this, we propose a pipeline for automatically extracting high-quality table-description pairs from the LaTeX sources of scientific articles. We study the effectiveness of state-of-the-art data-to-text generation models on SciGen and evaluate the results using common generation metrics and human evaluation. Our results and analyses show that adding high-quality unsupervised training data improves the correctness and reduces the hallucination in generated descriptions, however, the ability of state-of-the-art models is still severely limited on this task.[2]

## 1 Introduction

Data-to-text generation is one of the established tasks in NLP in which the input is structured data like tables and the output is a text that describes the data (Belz, 2008; Wiseman et al., 2017; Thomson et al., 2020; Lebret et al., 2016; Qader et al., 2018; Gardent et al., 2017). In this paper, we introduce SciGen, a new data-to-text generation dataset that contains pairs of scientific tables and their corresponding descriptions. A large number of table descriptions from the computer science domain require at least one type of arithmetic reasoning—e.g., argMax, argMin, comparison, subtraction, etc—over table values. This indicates that humans prefer to use reasoning to describe scientific tables.[3] Therefore, generative models that can describe scientific tables should be able to perform arithmetic reasoning.

In this direction, we created SciGen to enable the development and evaluation of generation models with arithmetic reasoning capabilities. For creating SciGen, we selected tables and their corresponding

---

*Contributions made prior to joining Amazon.

[2]Data, code, and human evaluations are available at `https://github.com/UKPLab/SciGen`.

[3]For instance, we randomly select 30 tables from different articles in this domain, and 60% of these tables were described by using arithmetic reasoning.

35th Conference on Neural Information Processing Systems (NeurIPS 2021) Track on Datasets and Benchmarks.

descriptions from the computer science articles with the following properties: (1) the tables mostly contain numerical values, and (2) the corresponding descriptions are the result of arithmetic reasoning over table values.

We release the dataset in three settings based on the size of the training data. The *few-shot* setting contains table-description pairs that are annotated by one of the authors of the corresponding articles. Since expert annotation does not scale to large data sizes, we introduce a pipeline consisting of automatic pairing and pruning techniques to extract high-quality table-description pairs in an unsupervised way. We extend the expert annotations in the few-shot setting using the automatically extracted pairs to create the medium and large splits of the dataset. All three settings use the same expert-annotated test data.

We study state-of-the-art data-to-text generation models—including BART (Lewis et al., 2020) and T5 (Raffel et al., 2020) pretrained language models—on SciGen. We evaluate the results using common generation evaluation metrics as well as human evaluation. Our results show that (1) while the generated outputs by the examined models are coherent and fluent and resemble valid descriptions, they are mostly factually incorrect due to the lack of arithmetic reasoning capabilities, (2) the addition of automatically extracted training data reduces the hallucination and improves the correctness of the generated descriptions. However, the extent that they can improve the correctness and arithmetic reasoning capabilities is limited, and (3) none of the common generation metrics can properly discriminate reasoning-aware text generation outputs. Since human evaluation is costly and time-consuming, developing proper automatic evaluation metrics is an important future direction to facilitate the progress of this task.

Overall, the main contributions of this paper are:

- SciGen, a reasoning-aware data-to-text generation dataset based on scientific articles.
- A pipeline for extracting table-description pairs from LaTeX files of scientific articles that provides high-quality unsupervised training data, and facilitates future annotation studies for new domains.
- Annotated data for human evaluations to facilitate creating new evaluation metrics.

## 2   Related Work

The task of data-to-text generation is to generate coherent, relevant, and meaningful natural language text that describes the non-linguistic input data like tables, knowledge bases, tuples, or graphs (Reiter and Dale, 2000; Gatt and Krahmer, 2018). Existing datasets for data-to-text generation cover various domains and applications including sport reports (Wiseman et al., 2017; Thomson et al., 2020; van der Lee et al., 2017), weather reports or forecast (Belz, 2008; Balakrishnan et al., 2019), restaurant descriptions (Dušek et al., 2020; Oraby et al., 2018; Reed et al., 2018), biographies (Lebret et al., 2016; Nema et al., 2018), entity descriptions (Qader et al., 2018; Wang et al., 2018), as well as open-domain datasets (Gardent et al., 2017; Parikh et al., 2020).

The textual descriptions in the majority of existing datasets mostly contain a verbalized summary of the content in the data, and are therefore surface-level summaries of the data (Chen and Mooney, 2008; Belz et al., 2011; Lebret et al., 2016; Gardent et al., 2017; Dušek et al., 2018; Koncel-Kedziorski et al., 2019; Nan et al., 2021; Parikh et al., 2020). SciGen, on the other hand, goes beyond the surface realization of the input data and requires arithmetic reasoning for text generation. The most related dataset to SciGen is LogicNLG (Chen et al., 2020a), in which the text generation step also requires logical reasoning.

LogicNLG is automatically created based on TabFact (Chen et al., 2020b), a table-based fact verification dataset. For creating TabFact, annotators were asked to write "refute" and "entailment" statements based on Wikipedia tables. The statements were classified into simple and complex: simple statements are verifiable without involving logical inference, and complex statements involve multiple rows of the tables as well as logical operations such as summary, argMax, argMin, count, comparison, average, etc. LogicNLG contains the complex statements of TabFact that are labeled as entailment given their corresponding table.

Apart from their domains—i.e., Wikipedia vs. scientific texts, there are two main differences between LogicNLG and SciGen. First, annotators of TabFact were asked to generate multiple statements

per table. As a result, each text only describes a part of the table—i.e., on average two rows of the table—and it often only contains one type of reasoning. The relevant rows of the table for each text are identified automatically in LogicNLG, and since identifying the relevant rows is not a trivial task, the LogicNLG examples are noisy. SciGen, on the other hand, only contains one description per table and it may contain multiple types of reasoning. SciGen is therefore more challenging than LogicNLG based on both data complexity and text complexity. For instance, LogicNLG descriptions contain 14 words on average, compared to 116 words in SciGen.[4]

Second, the types of logical operations that are used for creating TabFact, and therefore LogicNLG, are not limited to arithmetic operations. Based on Chen et al. (2020b)'s analyses, count is the most common logical operation in TabFact's complex statements. However, it also contains other types of reasoning like temporal reasoning, e.g., about 1200 descriptions in LogicNLG are generated based on before/after operations. SciGen, on the other hand, is designed for evaluating arithmetic reasoning.

Table 1 compares SciGen with recent table-to-text generation datasets based on various properties, in particular, (a) *data complexity* that is measured by the average number of cells in each table, (b) *text complexity* that is measured by the average number of words and the size of the vocabulary in the target text, and (c) the *reasoning* requirement to generate target texts.

| Dataset | Pairs | Cell | Num. | \|Text\| | \|Vocab\| | Domain | Annotation | Reasoning |
|---|---|---|---|---|---|---|---|---|
| WikiBIO | 400K | 17 | 3 | 97 | 400K | Biography | Automated | No |
| Rotowire | 11K | 649 | 429 | 337 | 11.3K | Basketball | Automated | Few |
| ToTTo | 136K | 3 | 1 | 17 | 136K | Open (Wikipedia) | Human | Few |
| LogicNLG | 37K | 91 | 35 | 14 | 122K | Open (Wikipedia) | Human/Automated | Yes |
| **SciGen** (few-shot) | 1.3K | 54 | 35 | 116 | 11K | Scientific | Expert | Yes |
| **SciGen** (medium) | 18K | 51 | 34 | 124 | 54K | Scientific | Expert/Automated | Yes |
| **SciGen** (Large) | 53K | 55 | 38 | 133 | 127K | Scientific | Expert/Automated | Yes |

Table 1: Comparison of SciGen to recent table-to-text generation datasets. *Pairs* shows the number of annotated pairs in each dataset. The *Cell* and *Num.* columns show the average number of total cells and the cells with numerical values in tables, respectively. \|*Text*\| reports the average numbers of words in descriptions. \|*Vocab*\| is the length of the corresponding vocabulary in each dataset.

## 3 Dataset and Problem Definition

### 3.1 Problem Definition

SciGen is a dataset for generating descriptions from scientific tables by reasoning over their content. An input in SciGen is a table $T$ extracted from a scientific article with its corresponding caption $C$, which is a word sequence containing one or few sentences about the table. $T = \{R_1, \ldots, R_n\}$ is represented as a list of lists that contains table rows. The task is to generate a textual description $D$ for the table that describes the most important findings of $T$ by reasoning over its content.

### 3.2 Expert Annotations

For creating SciGen, we have selected scientific articles from arXiv.org that are accompanied by their corresponding LaTeX sources. The selected articles are mainly from "Computation and Language" and "Machine Learning" fields of "Computer Science".

The expert annotations are done by one of the authors of the selected articles. We provide the authors a google drive page containing the PDF of their article. We ask the authors to (a) select the text spans in their article that describe a table, and (b) add the corresponding table number as a comment to each of the annotated spans.[5] Figure 1 shows a table from SciGen alongside its annotated description.[6] We ask authors to only annotate descriptions that can be generated from the table and its caption. This excludes descriptions that go beyond the table content, e.g., explaining the reason for performance differences in the non-highlighted text of Figure 1. We extract the annotated text

---

[4]In addition, descriptions of TabFact are generated by Amazon Mechanical Turkers while descriptions in SciGen are selected from the existing text in scientific articles, and therefore, they are more natural.

[5]All the participating authors in our annotation have explicitly agreed that their annotations will be included in a public dataset for research purposes.

[6]The table and description are from Moosavi et al. (2019).

spans and their comments from the PDF. We use the comments to pair the annotated descriptions with their corresponding tables. We extract tables using the AxCell tool (Kardas et al., 2020) from LaTeX sources as extracting tables from PDF files can be noisy (Hou et al., 2019).

|  | in-domain | out-of-domain | | |
|---|---|---|---|---|
|  | MultiNLI | SNLI | Glockner | SICK |
| MQAN | 72.30 | 60.91 | 41.82 | 53.95 |
| + coverage | **73.84** | **65.38** | **78.69** | **54.55** |
| ESIM (ELMO) | 80.04 | 68.70 | 60.21 | 51.37 |
| + coverage | **80.38** | **70.05** | **67.47** | **52.65** |

Table 2: Impact of using coverage for improving generalization across different datasets of the same task (NLI). All models are trained on MultiNLI.

Table 2 shows the performance for both systems for in-domain (the MultiNLI development set) as well as out-of-domain evaluations on SNLI, Glockner, and SICK datasets.

The results show that coverage information considerably improves the generalization of both examined models across various NLI datasets. The resulting cross-dataset improvements on the SNLI and Glockner datasets are larger than those on the SICK dataset. The reason is that the dataset creation process and therefore, the task formulation is similar in SNLI and MultiNLI, but they are different from SICK. In particular, in the neutral pairs

Figure 1: A sample expert table-description annotation. The annotation process contains marking the text spans from the article that (1) describes the table, and (2) can be generated using the table and its caption.

The annotations of this step contain all table-description pairs regardless of whether they require performing arithmetic reasoning. Therefore, two expert annotators—i.e., students from the Computer Science department with related training for the task—examine the resulting annotations from the first step. In this step, we remove pairs in which the description does not involve arithmetic reasoning—e.g., "The dataset statistics are summarized in Table 2"—, and incorrect annotations, e.g., pairs in which the table or the annotated PDF texts were not extracted correctly.[7]

Table 2 presents the statistics of the expert-annotated table-description pairs in SciGen. The table shows the statistics for different domains of the dataset, i.e., "Computation and Language", "Machine Learning", and "Others". The articles of the "Others" domain belong to various fields like "Computational Geometry" and "Distributed, Parallel, and Cluster Computing".

| Domain | Articles | Pairs | Cell | Num. | Cap. Sent | Cap. Word | Desc. Sent | Desc. Word |
|---|---|---|---|---|---|---|---|---|
| Computation and Language | 299 | 792 | 54.6 | 35.7 | 2.5 | 35.1 | 5.7 | 113.6 |
| Machine Learning | 191 | 410 | 49.9 | 32.1 | 2.4 | 32.5 | 6.0 | 117.2 |
| Others | 57 | 136 | 60.1 | 36.7 | 1.9 | 25.5 | 6.1 | 126.9 |

Table 2: The statistic of the expert-annotated data in SciGen. All domains consist of the "Computer Science" articles from arXiv.org. The *Articles* and *Pairs* columns show the number of annotated articles and tables in each domain, respectively. *Cell* shows the average number of cells in the annotated tables. *Num.* shows the average number of cells containing numerical values. *Cap. sent* and *Cap. word* show the average number of sentences and words in captions, respectively. *Desc. sent* and *Desc. word* report the average number of sentences and words in descriptions, respectively.

### 3.3 Extracting Automatic Annotations

Annotating descriptions of tables in scientific articles requires expert knowledge about the content of the article and is very time-consuming. Therefore, there is a limit to the amount of training data that can be created using expert annotations. In this section, we propose a pipeline to extract table-description pairs using the corresponding LaTeX files of scientific articles. Using LaTeX sources, we can easily locate paragraphs of the article that include a reference to a table. This way, we can collect an arbitrarily large number of table-description pairs from any scientific domain.

Our pipeline also contains a post-processing step to discard pairs in which (a) the description (potentially) does not reason over the table content. This step filters the data based on a set of heuristic rules that are based on the word patterns of the caption or the description[8] , (b) the table does not

---

[7]This step results in removing 970 table-descriptions.

[8]E.g., table-description pairs in which the caption contains word patterns like "details of dataset" or "summary of dataset" mostly presents the statistics of a dataset and their corresponding descriptions often do not contain arithmetic reasoning on table's content.

| | Pairs | \|Text\| | BLEU | METEOR | MScore | BertS | BLEURT |
|---|---|---|---|---|---|---|---|
| automatic annotations | 950 | 182 | 31.38 | 0.64 | 0.37 | 0.90 | -0.34 |
| +post-processing | 380 | 123 | 48.36 | 0.70 | 0.44 | 0.92 | -0.13 |

Table 3: Comparing the similarity of the automatically extracted table-description pairs, before and after post-processing, to the expert annotations based on BLEU, METEOR, MoverScore, BertScore, and BLEURT metrics. *Pairs* shows the number of common tables in the expert and automatically extracted annotations. *|Text|* shows the average number of words in descriptions.

contain numerical values, (c) the description describes multiple tables or figures, (d) the table is not extracted correctly by the Axcell tool, and (e) the description is too short—i.e., less than 15 words—or too long, i.e., longer than 400 words. Additionally, we shorten the paragraphs that consist of multiple subsections to only contain the one that relates to the target table.

To assess the quality of the resulting data, we automatically extract table-description pairs from the articles in our expert-annotated data and compare the automatically extracted descriptions with those that are manually annotated.

Based on our analysis, in 71% of the common examples between expert-annotated and automatic table-description pairs, the automatically extracted descriptions contain expert annotations. The average length of descriptions in the expert annotations of these examples is 95 words while it is 113 words for their corresponding automatic annotations.

There are also cases in which the automatic description does not contain the whole annotated description—i.e., in 29% of the common examples, the automatic description is shorter than the expert annotation. Such cases include expert annotations that spread over multiple paragraphs among which only one or few have a reference to the table, e.g., the annotation in the example of Figure 1.

Table 3 reports the similarity of automatic annotations, before and after post-processing, to expert annotations according to the evaluation metrics of § 4.3. For all these metrics, higher scores indicate higher similarity. As we see, the post-processing step improves the similarity of automatically extracted annotations with the expert annotations.

Our automatic table-description extraction pipeline (1) will make it possible to collect high-quality unsupervised table-description pairs from any new domain, for which we have access to LaTeX files, and (2) will facilitate expert annotations by suggesting related paragraphs to each table as well as identifying potential reasoning-aware descriptions. This way, annotators will not need to read the whole article or section for finding the descriptions.

**Limitation:** According to our analysis, automatically extracted annotations are often longer than their corresponding expert annotations. Therefore, they contain additional information that cannot be generated by only considering the table and its caption, e.g., describing the intuitions and reasons for the results of the tables. However, while the quality of the automatically extracted pairs is lower than our expert annotations, we show (§ 5.2) that their incorporation improves the quality of the generated descriptions.

### 3.4 Dataset Splits

We release the dataset in three different settings: (1) *few-shot*, (2) *medium*, and (3) *large*. The data splits in *few-shot* only contain table-description pairs from expert annotations. The training and development sets in this setting only contain pairs from the "Computation and Language" (C&L) articles. We split the test set into "C&L" and "Other" domains, in which the "Other" domain mainly contains examples from the "Machine Learning" (ML) articles.

The training and development sets in the *medium* setting contain those in *few-shot* plus automatically extracted pairs from additional "C&L" articles. Similarly, the training and development sets in the *large* setting contain those in *medium* in addition to automatically extracted pairs from additional "ML" articles. The test data is the same in all three settings. The "Other" test set is as an *out-of-domain* evaluation set for *few-shot* and *medium*. Table 4 reports the statistics of the three settings.

| Setting | Domain | Train | Dev | Test |
|---------|--------|-------|-----|------|
| Few-shot | C&L | 200 | 100 | 492 |
| | Others | 0 | 0 | 546 |
| Medium | C&L | 200+13407 | 100+3352 | 492 |
| | Other | 0 | 0 | 546 |
| Large | C&L | 200+13407 | 100+3352 | 492 |
| | Other | 26362 | 8677 | 546 |

Table 4: Number of table-description pairs in the training, development, and test sets.

# 4 Experimental Setup

## 4.1 Baselines

Motivated by the results of Ribeiro et al. (2021) that show the BART (Lewis et al., 2020) and T5 (Raffel et al., 2020) pretrained language models consistently outperform recent specialized data-to-text models on various benchmarks, we study the effectiveness of these two models on our dataset. For the BART baseline, we use the facebook/bart-large pre-trained model from HuggingFace's Transformers (Wolf et al., 2020) with 400M parameters. For the T5 model, we use the T5-base and T5-large models from HuggingFace's Transformers with 220M, and 770M parameters, respectively. All models are available at `https://github.com/huggingface/transformers/tree/v2.10`.

## 4.2 Input Representation

For using text-to-text generation baselines, we convert input tables into a text sequence. In order to preserve the structure of the table, we use three special tokens to specify the beginning of rows, cells, and the caption of the table, namely the "<R>", "<C>", and "<CAP>" tokens, respectively. Figure 2 shows an input table with its corresponding input representation.

|  | ellipsis (infl.) | ellipsis (VP) |
|--|------------------|---------------|
| baseline | 53.0 | 28.4 |
| concat | **76.2** | 76.6 |
| s-hier-to-2.tied | 66.4 | 65.6 |
| CADec | 72.2 | **80.0** |

Table 8: Accuracy on ellipsis test set.

<R> <C> [EMPTY] <C> [BOLD] ellipsis (infl.) <C> [BOLD] ellipsis (VP) <R> <C> baseline <C> 53.0 <C> 28.4 <R> <C> concat <C> [BOLD] 76.2 <C> 76.6 <R> <C> s-hier-to-2.tied <C> 66.4 <C> 65.6 <R> <C> CADec <C> 72.2 <C> [BOLD] 80.0 <CAP> Table 8: Accuracy on ellipsis test set.

Figure 2: Sample table from Voita et al. (2019) with its corresponding input representation. The "<R>", "<C>", and "<CAP>" special tokens specify the start of rows, cells, and captions, respectively.

## 4.3 Automatic Evaluation Metrics

**BLEU** (Papineni et al., 2002) is one of the most common evaluation metrics for text generation. It computes the geometric average of the precision over output text's n-grams. We use SacreBLEU (Post, 2018) that produces comparable and reproducible BLEU scores.

**METEOR** (Denkowski and Lavie, 2014) aligns the output text to the reference text and calculates sentence-level similarity scores for the alignments.

**BertScore** (Zhang et al., 2020) uses BERT embeddings and matches words in the output and reference sentences by cosine similarity. It then computes precision, recall, and $F_1$ measures based on the resulting matching.

**MoverScore** (Zhao et al., 2019) computes the distance between the contextual representation of the output and reference texts. It captures the amount of shared content between two texts as well

| $D_1$ | The results of Table 2 shows that the addition of coverage features improves the performance of MQAN by 1.54, 4.47, 36.87, and 0,6 points on MultiNLI, SNLI, Glockner, and SICK, respectively. Similarly, it improves the performance of ESIM (ELMO) by 0.34, 1.35, 7,26, and 1,28 on the mentioned datasets, respectively. We observe the highest improvements of both systems on the Glockner dataset. |
|---|---|
| $D_2$ | The results of using coverage for MQAN and ESIM (ELMO) systems on various datasets are reported in Table 2. The results show that the addition of coverage features significantly decrease the performance for both MQAN and ESIM (ELMO) baselines. We observe the highest drop in the Glockner dataset. |

Table 5: A correct ($D_1$) and an incorrect ($D_2$) description for the table in Figure 1.

as how much the output texts deviate from the reference. It uses BERT embeddings for computing contextualized representations.

**BLEURT** (Sellam et al., 2020) is a learned evaluation metric based on BERT. It is first pre-trained on synthetic examples and then fine-tuned on human judgments for the task of machine translation.

**Limitation:** The above metrics mostly measure the surface similarity of generated descriptions to gold ones, and they cannot evaluate the factual correctness of the generated descriptions given their corresponding tables. For instance, consider the sample descriptions in Table 5, where $D_1$ accurately describe the results while $D_2$ includes completely incorrect conclusions from the table. The BLEU, METEOR, BertScore, MoverScore, and BLEURT scores for $D_1$ compared to the gold description (marked in Figure 1) are 11.65, 0.35, 0.86, 0.27, and -0.57, respectively. These scores for $D_2$ are 12.18, 0.30, 0.87, 0.30, and -0.54, respectively. As we see, results for both descriptions based on all these evaluation metrics are in the same range, and in some cases higher for the incorrect description.

# 5 Results

## 5.1 Automatic Evaluation

Table 6 shows the results of our baselines—i.e., BART-large, T5-base, and T5-large—on different splits of the SciGen dataset using the evaluation metrics of §4.3. We also report the results for the zero-shot setting in which the models are evaluated on the test split without any fine-tuning.

Based on the results: (1) except for BertScore, the value range for the rest of the metrics is very low, (2) BertScore values are very high for all the experiments, however, as we see in § 5.2, generated descriptions are far from being accurate,[9] (3) there is not a clear agreement between the rankings of different metrics for the examined models and settings, and (4) according to automatic metrics BART performs better than the two other models, however, as we see in § 5.2, T5-large performs on-par with or in cases better than BART according to human evaluation.

These results indicate that automatic metrics are not sufficient for this task. As an example of model outputs, Table 7 shows the outputs of the BART-large model on one of the "C&L" test examples, i.e., the table in Figure 2.

## 5.2 Human Evaluation

For human evaluation, we select 58 table-description pairs from the SciGen "C&L" test set and their corresponding system-generated descriptions from the BART and T5-large models for the three settings. We break down each description, both gold and system-generated ones—i.e., $58 \times 2 \times 3$ descriptions–to a list of individual statements. For instance, the corresponding statements with the gold description in Table 7 are (a) "For ellipsis, both models improve substantially over the baseline (by 19-51 percentage points)", (b) "concat is stronger for inflection tasks", and (c) "CADec is stronger for VPellipsis".

We assign one of the following labels to each of the extracted statements from system-generated descriptions:

- *Entailed*: a generated statement that is entailed by the corresponding gold description, i.e., is equivalent to one of the extracted statements from the gold description.

---

[9] The output of models in *zero-shot* is non-sensible, and yet it receives 0.74 BertScore.

| Setting | Model | BLEU | METEOR | MoverS | BertS | BLEURT |
|---|---|---|---|---|---|---|
| | | | Test(C&L) | | | |
| | BART | 1.11 | 0.11 | -0.02 | 0.74 | -1.23 |
| Zero | T5-base | 0.69 | 0.04 | -0.05 | 0.76 | -1.31 |
| | T5-large | 1.16 | 0.06 | -0.06 | 0.76 | -1.28 |
| | BART | 4.73 | 0.22 | **0.14** | **0.84** | **-0.66** |
| Few | T5-base | 2.59 | 0.13 | 0.02 | 0.79 | -1.09 |
| | T5-large | 3.16 | 0.16 | 0.06 | 0.81 | -0.95 |
| | BART | **5.30** | **0.23** | 0.13 | **0.84** | -0.72 |
| Medium | T5-base | 3.32 | 0.15 | 0.05 | 0.82 | -0.89 |
| | T5-large | 3.65 | 0.17 | 0.10 | 0.83 | -0.77 |
| | BART | 5.04 | 0.22 | **0.14** | **0.84** | -0.71 |
| Large | T5-base | 3.38 | 0.15 | 0.06 | 0.82 | -0.85 |
| | T5-large | 3.84 | 0.18 | 0.10 | 0.83 | -0.79 |
| | | | Test(Other) | | | |
| | BART | 0.92 | 0.09 | -0.03 | 0.74 | -1.29 |
| Zero | T5-base | 0.86 | 0.04 | -0.05 | 0.76 | -1.24 |
| | T5-large | 1.26 | 0.06 | -0.05 | 0.76 | -1.24 |
| | BART | 4.26 | **0.22** | **0.13** | **0.84** | **-0.66** |
| Few | T5-base | 2.12 | 0.11 | 0.00 | 0.78 | -1.13 |
| | T5-large | 2.58 | 0.14 | 0.03 | 0.81 | -1.01 |
| | BART | 4.64 | **0.22** | 0.12 | **0.84** | -0.71 |
| Medium | T5-base | 3.17 | 0.14 | 0.04 | 0.81 | -0.91 |
| | T5-large | 3.00 | 0.16 | 0.07 | 0.82 | -0.81 |
| | BART | **4.66** | 0.20 | 0.12 | **0.84** | -0.72 |
| Large | T5-base | 3.66 | 0.14 | 0.05 | 0.81 | -0.86 |
| | T5-large | 3.61 | 0.17 | 0.07 | 0.82 | -0.81 |

Table 6: Performance of the BART-large, T5-base, and T5-large models on the *few-shot*, *Medium*, and *Large* settings based on BLEU, METEOR, MoverScore, BertScore, and BLEURT metrics. *Zero* rows report the zero-shot performance of the examined models without finetuning on SciGen. We report the scores on "C&L" and "Other" test domains separately. The "Other" examples are out-of-domain test data for the *few-shot* and *Medium* settings. Highest scores for each domain are boldfaced.

| Setting | Description |
|---|---|
| Gold | For ellipsis, both models improve substantially over the baseline (by 19-51 percentage points), with concat stronger for inflection tasks and CADec stronger for VPellipsis. |
| Few | table 8 shows the performance of our model on the ellipsis test set. **our concat model outperforms all the other models that do not use concat embeddings.** |
| Med. | the results show that both concat and cadec significantly outperform the baseline and s-hier-to-2.tied on the ellipsis test set. the concat model is better than the baseline on both inflected and verb-inflated forms, **while** cadec is better on the vp version. on the inflated version, cadec outperforms the baseline by a large margin and **is within 3 points of the best model.** the model with the best accuracy on the verb portion of the test set (80.0%) is very close to the best performance on the noun portion (75.6%). **the results indicate that cadec has better generalizability than the other models**. |
| Large | the concatenation model concatenates the output of the concatenated encoder and decoder, and **the s-hier-to-2.tied model performs slightly better than concat on both ellipsis (infl.) and vice versa**. cadec outperforms the other two models by a large margin, achieving 80.0% accuracy on the vp ellipsoids, which is the new state of the art. |

Table 7: Sample outputs from the BART model on few-shot, medium, and large settings on the corresponding table from Figure 2. Factually correct statements are marked with green, factually incorrect statements and hallucinations are marked with **red** and blue, respectively.

- *Extra*: a statement that is not entailed by the gold description but is correct based on the table content.
- *Incorrect*: a statement that is relevant to the table but is factually incorrect. For instance, "the s-hier-to-2.tied model performs slightly better than concat on both ellipsis (infl.) and vice versa." in Table 7 contains relevant entities that are mentioned in the table, but the statement is incorrect.
- *Hallucinated*: a statement that is unrelated to the table.

We then compute four metrics as follows:

1. *Recall:* the ratio of the statements in the gold description that are covered by the system-generated description—i.e., $\frac{|\text{entailed statements}|}{|\text{gold statements}|}$ per description.

2. *Precision:* the ratio of the statements in the system-generated description that exist in the gold description—i.e., $\frac{|\text{entailed statements}|}{|\text{generated statements}|}$ per description.

3. *Correctness:* the ratio of the statements in the system-generated description that are factually correct—i.e., $\frac{|\text{entailed statements}|+|\text{extra statements}|}{|\text{generated statements}|}$.

4. *Hallucination:* the ratio of irrelevant statements with regard to the table that is computed as $\frac{|\text{hallucinated statements}|}{|\text{generated statements}|}$

| Model | Setting | Recall | Precision | Correctness | Hallucination |
|-------|---------|--------|-----------|-------------|---------------|
| | Few | 0.04 | 0.04 | 0.10 | 0.44 |
| BART | Medium | **0.13** | 0.10 | **0.39** | 0.23 |
| | Large | 0.08 | 0.06 | 0.22 | 0.28 |
| | Few | 0.04 | 0.03 | 0.08 | 0.26 |
| T5-Large | Medium | 0.11 | 0.11 | **0.39** | 0.22 |
| | Large | 0.10 | **0.13** | 0.35 | **0.14** |

Table 8: The results of human evaluation for the BART and T5-large models according to the recall, precision, correctness, and hallucination metrics.

Table 8 presents the results of the human evaluation. First, the addition of automatically extracted pairs in the medium and large settings improves the recall, precision, and correctness of the generated descriptions and decreases their hallucination.

Second, compared to the medium setting, the generated descriptions in the large setting contain a larger number of *factually-incorrect* facts, i.e., lower correctness. This could be due to the fact that the additional table-description pairs in large are from a different domain, i.e., "ML".

Third, BART received a higher score compared to T5-large based on all automatic metrics in Table 6. However, they both receive on-par scores according to human evaluation while T5-large generated descriptions contain less hallucination. The generated descriptions by BART are generally longer than those by T5, e.g., the average number of generated words per description for the *few-shot*, *medium*, and *large* settings by BART and T5-large are 84, 118, 98 and 72, 91, 91, respectively.

Finally, there is still a large gap to solve for the SciGen dataset, i.e., in the best case, only 39% of the generated statements are correct and the ratio of incorrect statements is still very high for all the examined outputs. Incorrect statements are those in which a wrong conclusion is made based on the table content, which indicates the lack of arithmetic reasoning in the examined models. Also, the recall and precision values are lower than those of correctness indicating that apart from arithmetic reasoning, the examined models also do not perform well for relevant content selection.

## 6 Broader Impact

Enabling reasoning-aware data-to-text generation from scientific tables can assist scientists by generating the result section of their articles using tables of experimental results, or practitioners by generating the explanation of their result tables. In addition, it helps to develop specialized chatbots that can generate answers based on tables' contents. However, depending on the quality of the resulting model, there are risks concerning generating incorrect descriptions that contain disinformation regarding the presented results in the table. By developing models with better reasoning capabilities, the factuality and correctness of the generated descriptions will also improve. Another risk is that such models can be used for generating fake scientific articles. At the moment, common NLP models do not perform very well on scientific domains. As they become better, customizing models like Grover (Zellers et al., 2019) for detecting fake scientific content can be used to mitigate this risk.

# 7 Conclusions

We introduce SciGen that is a challenging dataset for reasoning-aware data-to-text generation. The input data in SciGen are tables from scientific articles and generating their corresponding descriptions requires arithmetic reasoning over table values. Annotating scientific articles is costly and time-consuming and does not scale to large data sizes. To tackle that, we provide a pipeline to extract high-quality unsupervised table-description pairs from the corresponding LaTeX files of scientific articles. We release SciGen in three different settings—few-shot, medium, and large—based on the size of the training data. The few-shot setting and the test set contain expert-annotated pairs while the training data in medium and large settings contain automatically extracted pairs. We study state-of-the-art data-to-text generation models on SciGen and evaluate the results using common generation evaluation metrics as well as human evaluation. Our results show that (1) common metrics are not suitable for evaluating reasoning-aware text generation—i.e., they do not correlate with human evaluation and they also do not agree with each other—, and (2) adding automatically extracted annotations improves the correctness of the generated descriptions and reduces the hallucination. However, there is still a large gap to solve for this dataset.

## Acknowledgements

The authors would like to thank Xia Zeng, Dennis Kuhn, Ragini Menon, and Gisela Vallejo for their great efforts in the data collection process. We gratefully acknowledge the help of numerous members of this research community in helping us with the annotations. This work was possible thanks to all these contributing researchers. We would also like to thank Michael Bugert, Jan-Christoph Kile, Ji-Ung Lee, Yevgeniy Puzikov, Kevin Stowe, and Ajie Utama for their valuable feedbacks on the paper. This work has been supported by the German Research Foundation (DFG) as part of the QASciInf project (grant GU 798/18-3), and the German Federal Ministry of Education and Research and the Hessian Ministry of Higher Education, Research, Science and the Arts within their joint support of the National Research Center for Applied Cybersecurity ATHENE.

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
