# OpenReview forum: "SciGen: a Dataset for Reasoning-Aware Text Generation from Scientific Tables"
_NeurIPS.cc/2021/Track/Datasets_and_Benchmarks/Round2 — NeurIPS 2021 Datasets and Benchmarks Track (Round 2)_

### Official Review · Reviewer_dDP4 · 2021-09-19
**A challenging dataset for scientific table-to-text generation**

**Rating:** 7
**Confidence:** 3
**Clarity:** The paper is easy to read and follow.

**Strengths:**

• This paper proposed a much more challenging dataset for table-to-text generation in terms of data complexity and text complexity. Three different settings are further provided.
•  Apart from expert annotated data, a pipeline method is proposed to extract table-description pairs from LaTeX files of scientific articles, which may facilitate future annotation studies for new domains.
• The data construction process seem thoughtful and reasonable.

**Weaknesses:**

My biggest concern is how to evaluate the models on this dataset. It seems to me that the automatic evaluations metrics used in this paper are not suitable for such a reasoning-aware dataset. These metrics are not reliable as they do not agree with each other (including the human evaluation). For example, I am wondering why BERTScore is so high for different settings.

The most challenging problem of SciGen is how to evaluate the ability to perform arithmetic reasoning over the table, which is also the core problem of our paper. Other automatic metrics should be designed to evaluate models trained on the dataset. For example, prior effort LogicNLG used three additional metrics apart from perplexity and BLEU score to evaluate the logical fidelity of the generated sentences, including parsing-based, NLI-based and adversarial metrics. Are these evaluation metrics applicable for SciGen? Solely relying on human evaluation may hinder researchers from developing models for the dataset, as the human evaluations sometimes are not consistent and reliable as well.

**Additional Feedback:**

No

**Correctness:**

The construction of the dataset and baselines are sound, though I do have concerns related to the evaluation metrics mentioned above.

**Documentation:**

The dataset collection process is well-documented (in the supplementary materials) and a URL to the dataset is also provided.

**Ethics:**

No ethical concerns.

**Relation To Prior Work:**

The related work section gave a detailed comparison between Scigen and LogicNLG, in which the text generation also requires logical reasoning.

**Summary And Contributions:**

This paper introduced a challenging dataset SciGen for scientific table-to-text generation. Generating the textual descriptions from their corresponding tables requires arithmetic reasoning over table values.  SciGen has three different settings including few-shot, medium, and large. The few-shot setting and the test set contain expert-annotated pairs. For constructing large-scale training data, they further developed a pipeline to extract table-description pairs from the corresponding latex files of scientific articles. The paper evaluates BART and T5 on the proposed datasets by using automatic metrics and manual efforts.

---

> ### Author Response · Authors · 2021-09-25
> **Evaluation challenges**
>
> Thanks for your valuable and positive feedback.
>
>
> (1) My biggest concern is how to evaluate the models on this dataset. It seems to me that the automatic evaluations metrics  used in this paper are not suitable for such a reasoning-aware dataset. These metrics are not reliable as they do not agree with each other (including the human evaluation). For example, I am wondering why BERTScore is so high for different settings.
>
> - There are numerous work that show the unsuitability of automatic measures, and therefore the necessity for including human judgments, for evaluating NLG systems (e.g., Ananthakrishnan et al., 2007; Novikova et al., 2017; Sulem et al., 2018; Reiter, 2018, , Deutsch et al., 2021) even on standard text generation datasets. From the evaluation perspective, SciGen provides a great evaluation case for developing more reliable evaluation metrics for novel domains and challenging generation tasks.
> - Regarding the high values of BertScore, our hypothesis is that it is due to the large number of unknown tokens in both gold and generated descriptions.
>
> (2) The most challenging problem of SciGen is how to evaluate the ability to perform arithmetic reasoning over the table, which is also the core problem of our paper. Other automatic metrics should be designed to evaluate models trained on the dataset. For example, prior effort LogicNLG used three additional metrics apart from perplexity and BLEU score to evaluate the logical fidelity of the generated sentences, including parsing-based, NLI-based and adversarial metrics. Are these evaluation metrics applicable for SciGen?
>
> - We considered evaluating these three metrics used in LogicNLG. The parser that is used in “Parsing-based Evaluation” and the NLI model in “NLI-based Evaluation” require a training step. They are trained on LogicNLG and TabFact datasets. Their outputs on new domains are noisy and errourness and cannot be used for evaluation purposes. For their adversarial evaluation, Chen et al used human workers from Amazon Mechanical Turk. LogicNLG descriptions are shorter and from a general domain, so their evaluation is easier for MTurk while it is not the case for SciGen.
>
> R. Ananthakrishnan, P. Bhattacharyya, M. Sasikumar, R.M. Shah. Some Issues in Automatic Evaluation of English-Hindi MT: More Blues for BLEU, ICON (2007)
>
> J. Novikova, O. Dušek, A.C. Curry, V. Rieser. Why we need new evaluation metrics for NLG. Proceedings of the 2017 Conference on Empirical Methods in Natural Language Processing (2017), pp. 2241-2252, 10.18653/v1/D17-1237
>
> E. Reiter. A structured review of the validity of BLEU. Comput. Linguist. (2018), pp. 1-12, 10.1162/coli_a_00322
>
> D. Deutsch, R. Dror, and D. Roth. A Statistical Analysis of Summarization Evaluation Metrics Using Resampling Methods. Transactions of the Association for Computational Linguistics (2021)

---

> > ### Comment · Reviewer_dDP4 · 2021-10-03
> > **Review Update**
> >
> > Thanks for the response, which addressed some of my concerns (evaluation metrics used in LogicNLG). However, I am not fully convinced by the response to my first concern. Of course, I know the unsuitability of automatic measures. My point is solely relying on human evaluation may hinder researchers from developing models for the dataset, as the human evaluations may not be consistent and reliable either. I would keep my rating for this paper.

---

### Official Review · Reviewer_wzAJ · 2021-09-20
**An interesting and very challenging new dataset for data-to-text generation**

**Rating:** 7
**Confidence:** 3
**Clarity:** The paper flows nicely and is easy to…

**Strengths:**

1. The dataset is very challenging while also being reasonably objective (from a human perspective) in terms of whether a description is correct or incorrect due to the focus on arithmetic reasoning.

2. Discussion of the validity of automatic evaluation metrics is useful and will hopefully lead to improved metrics being developed in future.

3. The usage of automatically extracted data to supplement the original training data is useful and will enable the size of the dataset to naturally grow over time as new papers are published.

**Weaknesses:**

[Update: The authors have provided a reply to the below concerns which addresses them. Because of this my score has been increased.]

1. The dataset is challenging to the point of it being hard to identify if any notable progress has been made in solving it. This is partially due to the lack of baselines. While T5 and BART were benchmarked, it would have been useful to benchmark older models to get a perspective on if the newer models are improving on this task.

2. An ablation on using even fewer than 200 training samples would have been useful to identify how rapidly performance improves with access to high quality expert descriptions.


**Additional Feedback:**

Given that evaluation with automatic metrics seem not sufficient, it appears difficult for future models to incorporate this dataset into their analysis without significant resource commitment in terms of expert human scorers (plus this opens up concerns of the scorers being different from those who scored the models in this work). This may be a problem for the general dataset-to-text generation domain, but I'm curious what the authors think of the difficulty of dataset adoption and evaluation.

**Correctness:**

The claims appear to be correct and the dataset seems to be soundly designed and constructed.

**Documentation:**

The documentation of how the dataset was collected is clear.

**Ethics:**

I don't see any notable ethical concerns with the dataset.

**Relation To Prior Work:**

The paper mentions prior work such as LogicNLG and provides comparisons that indicate how SciGen significantly differs from prior datasets.

**Summary And Contributions:**

The paper introduces SciGen, a new data-to-text generation dataset for generating arithmetic aware descriptions of scientific tables. SciGen is compared to prior work in the domain, most closely with LogicNLG, and describes how the increased length of the text descriptions and focus on arithmetic reasoning significantly increase the complexity of the problem. The paper compares BART and T5 performance to human baseline on SciGen, showing that ML models struggle heavily on the dataset and are very often producing wrong and factually incorrect statements. Furthermore, it is observed that common automated evaluation methods are not sufficient to differentiate the quality of generated descriptions.

---

> ### Author Response · Authors · 2021-09-25
> **Older baselines, and ablation study**
>
> Thanks for your valuable feedback.
>
> (1) The dataset is challenging to the point of it being hard to identify if any notable progress has been made in solving it. This is partially due to the lack of baselines. While T5 and BART were benchmarked, it would have been useful to benchmark older models to get a perspective on if the newer models are improving on this task.
>
> - The comparison of older non-pretrained and recent pretrained models is explored in the LogicNLG dataset by Chen et al. Based on their findings, compared to previous generation models, pretrained models generate more natural descriptions: “since we had a limited training set with a broader vocabulary, the pointer-generator model tended to result in repetitive words and failed to generate well-described descriptions. The pre-trained models, GPT2 and T5, generated more natural descriptions. While several pieces of text generated by GPT2 included numerical facts, they used numbers that were not extracted from table contents. The T5 models produced descriptions that were more related to table contents than GPT2.” We believe that the same holds for SciGen.
>
>
> (2) An ablation on using even fewer than 200 training samples would have been useful to identify how rapidly performance improves with access to high quality expert descriptions.
>
> - We run the BART models with 0 (zero-shot), 50, and 100 training examples and below are the automatic metrics:
>
> Zero-shot:        BLEURT (-1.397)  BLEU (0.512)  MoverScore (0.482) BertScore (0.743) Meteor (0.085)
> 50 examples:   BLEURT (-0.58)    BLEU (5.655)  MoverScore (0.535) BertScore (0.847) Meteor (0.228)
> 100 examples: BLEURT (-0.712)  BLEU (2.779)  MoverScore (0.529) BertScore (0.848) Meteor (0.182)
> We will add these results in the next revision.
>
> (3) Given that evaluation with automatic metrics seem not sufficient, it appears difficult for future models to incorporate this dataset into their analysis without significant resource commitment in terms of expert human scorers (plus this opens up concerns of the scorers being different from those who scored the models in this work). This may be a problem for the general dataset-to-text generation domain, but I'm curious what the authors think of the difficulty of dataset adoption and evaluation.
>
> - Recently there is a growing interest in evaluating generation evaluation metrics and proposing better alternatives. However, such developments and analyses are mainly focused on general text domains like the datasets from WMT shared tasks. We believe that it is important to develop metrics that apply to more novel text domains, like scientific texts, and more challenging generation tasks like reasoning-aware text generation. Our dataset and paper do not solve the problem of evaluating NLG systems on more challenging text domains and tasks, but it enables more research in this direction.

---

> > ### Comment · Reviewer_wzAJ · 2021-10-04
> > **Response**
> >
> > Thanks for the detailed reply. This has addressed the majority of my concerns and questions, and therefore I will increase my score.

---

### Official Review · Reviewer_d24g · 2021-09-20
**Interesting dataset for table to text generation in scientific domain**

**Rating:** 7
**Confidence:** 4
**Correctness:** seems good.

**Strengths:**

- An interesting large scale  table to text dataset for scientific domain.
- Three settings of training data (which also includes few shot setup)
- automatic evaluation of baseline techniques on 5 metrics shows large gap between human performance and baseline models which will enable future research



**Weaknesses:**

- How does the baseline model perform in zero shot setup? say the model trained on standard table to text dataset and tested on scigen. How does SOTA model perform in this setup is missing from the paper.
- The dataset is specific to particular topic/domain and the model trained on this dataset might not be generalisable to other topics/domains.


**Additional Feedback:**

- Have a zero shot baseline setup.


**Clarity:**

The paper needs to be correctly structured and overall writing needs some improvement.

**Documentation:**

Yes, data collection and annotation details are provided.

**Ethics:**

No ethical concerns.

**Relation To Prior Work:**

Yes discussed.

**Summary And Contributions:**

A new dataset from scientific domain with tables and corresponding textual description. Authors claim that SciGen has two unique properties 1) mostly contain tables with numerical values 2) Generating description require numerical reasoning over table values. To generate this dataset authors propose a technique to automatically extract table text pairs from the latex sources of scientific articles. So, the main contributions of this paper are two fold a) A new table to text dataset from scientific domain with tables containing mostly numeric values and require numerical reasoning to generate text b) a new technique to extract table text pair from latex sources of scientific articles.

---

> ### Author Response · Authors · 2021-09-25
> **Zero-shot setting baseline**
>
> Thanks for your valuable and positive feedback.
>
> (1) How does the baseline model perform in zero shot setup? say the model trained on standard table to text dataset and tested on scigen. How does SOTA model perform in this setup is missing from the paper.
>
> - Thanks for the suggestion. Below you can see the results of the BART-large model in the zero-shot setting
> BLEURT (-1.397)  BLEU (0.512)  MoverScore (0.482) BertScore (0.743) Meteor (0.085)
>
> All the generated outputs for this setting are nonsensical, e.g., “2525 / " " " / " : " " " " " : : " / : " : " " " : : : " : " " " : " " " : 11 : " 11 " 2 " " 10 " " " 1 " " 2 1 " 1 1 " v " " , " 1 " 2 " 1 " " , , m11 : 0 000 0 0 . . . 0 0 0 . 390 0 . 61 0 . 39 0 . 40 0 . 11 0 . 50 0 . 67 0 0 0 10 0 . 59 0 . 39 0 . 62 39 0 public 0 . 41 39 to 0 . 60 600 0 . 25 39 then 0 . 66 0 . 15 0 . 2 . 39 4 0 . 24 0 . 3 . 11 4 . 39 . 39 public 10 . 11 10 . 50 600 5 . 61 all tw 0 . 42 heat 0 . 5 . 11 5 . 40 50 4 . 11 . 2 : 2 . 11 11 . 11 6 . 61 10 . 61 4 . 5 to 1 . 11 9 . 11 2 . 2 to 10 . 2 6 . 11 1 . 2 5 . 11 2 . 11 15 . 11 to the best 10 . 40 4 . 2 4 . 40 . 2 . 40 some of the best 4 . 15 4 . 62 all 2 half 2 legal half 1 legal legal half 0 . 47 10 . 3 to the 4 . 3 of the 10 . 5 . . . 4 . 11 . . . 6 . 2 . . . 11 . 11 : 4 . 62 . 39 magn 2 . 5 - 4 . 5 : 1 . 11 3 . 11 8 . 11 16 . 11 7 . 2 10 . 62 10 . 39 light 6 . 11 am 0 . 4 . 39 hands to back to”
>
> (2) The dataset is specific to particular topic/domain and the model trained on this dataset might not be generalisable to other topics/domains.
>
> - This is a good point. In our experiments, we have included an out-of-domain evaluation on the "Test(other)" domain including pairs from the ML articles for the models trained on few-shot and medium settings (on CL articles). We didn't observe a big difference between in-domain and out-of-domain evaluations based on automatic evaluation metrics. Investigating transfer issues including transferring to domains other than computer science is left as future work.

---

### Official Review · Reviewer_5AJD · 2021-09-21
**First data-to-text dataset on scientific domain**

**Rating:** 7
**Confidence:** 3
**Correctness:** The paper is clear and concise.
**Clarity:** The paper is well written.

**Strengths:**

- First dataset to work on scientific tables specifically from the computer science domain.
- There aren't much of reasoning-aware table-to-text dataset and this dataset will add value the subtopic.

**Weaknesses:**

- The dataset focuses only on tables in computer science domain.
- Human annotation is there only for a small number of instances (1338). This happened because the authors of the documents were asked to annotates the parts of the document where tables are explained and it would be hard to get many authors to do this. Is it necessary to get the original authors to annotate? Getting non-author annotators might have been easier and they might have done more annotations with almost as accurate as the authors. As later in the paper it is shown that even the automated annotator is doing a decent job in identifying the descriptions of tables from document.
- The applications of the dataset is little. Model trained on these dataset may only work on tables from ML and CL papers. However, it would be interesting to see if the automated table-description pair extractor can broaden the scope by creating datasets for other domains.

**Additional Feedback:**

In CS fields, especially ML and CL, the bold letters are often used to show the best performer in the results table. It might not be the case with other domains. It would be interesting to see without the [BOLD] tag how the performance drops, and if it does how generalizable is dataset to another domain where bold letters aren't the norm.

**Documentation:**

The authors provide adequate details on the datasets and the evaluation setup.

**Ethics:**

The data is obtained from scientific documents publicly available, so there should be no ethical concerns.

**Relation To Prior Work:**

In relation to previous works, the ideas were clearly communicated.

**Summary And Contributions:**

The paper introduces a first of a kind dataset for data-to-text generation task on "scientific tables". The dataset was created using computer science documents in arXiv mostly from "Computation and Language","Machine Learning" domains. From these documents, tables and parts of the text where the tables are explained is extracted to create the data-to-text instances. This annotation of which part of text explains the table is either done manually by the original authors of these documents or by a rule-based automated system. The annotated data which does not contain any arithmetical reasoning was removed. Initial experiments we conducted on these datasets.

Contributions:
- A new reasoning-aware table-to-text dataset on research papers in computer science field.
- An automated system to extract table-description pairs from LaTeX files of documents.
- Initial experiments to show the difficulty of the task.

---

> ### Author Response · Authors · 2021-09-25
> **Response to Reviewer 5AJD**
>
> Thanks for your valuable and positive feedback.
>
> (1)  Is it necessary to get the original authors to annotate?
>
> - The annotators do not need to be the authors of articles as long as they are domain experts that are familiar with the topic of those scientific articles to accurately mark the descriptions. The main difference would be that the annotation time, and therefore cost, by non-authors will be high since they need to read and understand more parts of the article. As mentioned in the paper, our annotation extraction pipeline can be used for reducing this annotation time by suggesting related paragraphs of each table as well as identifying potential reasoning-aware descriptions.
>
> (2) The applications of the dataset is little. Model trained on these dataset may only work on tables from ML and CL papers. However, it would be interesting to see if the automated table-description pair extractor can broaden the scope by creating datasets for other domains. In CS fields, especially ML and CL, the bold letters are often used to show the best performer in the results table. It might not be the case with other domains. It would be interesting to see without the [BOLD] tag how the performance drops, and if it does how generalizable is dataset to another domain where bold letters aren't the norm.
>
> - This is an interesting point. We analyzed SciGen concerning the inclusion of [BOLD] tags in tables. The ratio of tables that do not contain any boldfaced numbers in SciGen is as follows: test (43.5%), few-shot training (41.5%), medium training (41.6%), large training (47.1%). As we see from these ratios, there is a reasonable ratio of tables without [BOLD] tags in both training and test splits. Therefore,  models that mainly learn to rely on these [BOLD] tags will not perform well on all the instances of SciGen. We believe that our data collection method is applicable for articles from other scientific domains. Extending SciGen with more diverse scientific domains is a very valuable future direction for which our paper lies the ground work.

---

### Decision · Program_Chairs · 2021-10-09

**Decision:**

Accept

**Comment:**

This paper proposes a new dataset for generating texts that describe tables in CS articles. There was broad consensus among the reviewers that this work is interesting and contributes to the field of data-to-text generation in a new and challenging domain that requires performing reasoning over the tables.